SOFTWARE

# mdciao: Accessible Analysis and Visualization of Molecular Dynamics Simulation Data

**Guillermo Pérez-Hernández**[ID][1][*], **Peter W. Hildebrand**[2,3,4]

**1** Charité – Universitätsmedizin Berlin, corporate member of Freie Universität Berlin and Humboldt-Universität zu Berlin, Institute of Medical Physics and Biophysics, Berlin, Germany, **2** Universität Leipzig, Medizinische Fakultät, Institut für Medizinische Physik und Biophysik, Leipzig, Germany, **3** Berlin Institute of Health at Charité – Universitätsmedizin Berlin, Berlin, Germany, **4** Center for Scalable Data Analytics and Artificial Intelligence (ScaDS.AI), Leipzig, Germany

* guillermo.perez@charite.de

## ABSTRACT

We present mdciao, an open-source command line tool and Python Application-Programming-Interface (API) for easy, one-shot analysis and representation of molecular dynamics (MD) simulation data. Building upon the widely used concept of residue-residue contact-frequencies, mdciao offers a wide spectrum of further analysis and representations, enriched with available domain specific annotations. The user-friendly interface offers pre-packaged solutions for non-expert users, while keeping customizability for expert ones. Emphasis has been put into automatically producing annotated, production-ready figures and tables. Furthermore, seamless on-the-fly query and inclusion of domain-specific generic residue numbering for GPCRs, GAIN-domains, G-proteins, and kinases is made possible through online lookups. This allows for easy selection and comparison across different systems, regardless of sequence identity, target residues or domains. Finally, the fully documented Python API allows users to include the basic or advanced mdciao functions in their analysis workflows, and provides numerous examples and Jupyter Notebook Tutorials. The source code is published under the GNU Lesser General Public License v3.0 or later and hosted on https://github.com/gph82/mdciao, and the documentation, including guides and examples, can be found at https://www.mdciao.org

## Introduction

Molecular Dynamics (MD) simulations are a widely used tool for the theoretical investigation of the dynamics of (bio)molecular systems with atomic-level detail [1].

In recent years, MD simulation tools have become increasingly user-friendly, and the hardware on which they run has become faster and cheaper [2,3]. Also, in addition to a growing number of dedicated MD trajectory repositories (e.g., [4,5]), machine-learning methods predicting structures[6–8] and trajectories [9,10] in one shot, have lowered the barrier to access ensembles and structures even more (for a recent review of MD data generally available online see [11]). Thus, side-stepping the question about how the data were generated and the accuracy of the underlying physical models, the challenge that non-expert simulators face shifts slowly from generating structural data (typically, MD trajectories) to analyzing and

**Data availability statement:** All relevant data are within the manuscript and its Supporting Information files. mdciao is published under the GNU Lesser General Public License v3.0 or later. The source code is hosted on https://github.com/gph82/mdciao, the current stable release is hosted at https://pypi.org/project/mdciao/ and the documentation, including guides and examples can be found at https://proteinformatics.uni-leipzig.de/mdciao. The release used for this manuscript is v.0.0.9.

**Funding:** This work was supported by Deutsche Forschungsgemeinschaft (German Research Foundation) (to PWH) through SFB1423, project 421152132, subproject C01 and Z04 and the Stiftung Charité (to PWH). The funders had no role in study design, data collection and analysis, decision to publish, or preparation of the manuscript.

**Competing interests:** The authors have declared that no competing interests exist.

summarizing it. The depth and scope of this analysis can range from fairly straightforward and intuition-guided to arbitrarily complex and automated [12].

Many software solutions have been produced over the last decades to analyze MD data, offering different degrees of pre-packaged solutions to experts and non-experts. Usually, first and easiest step is to visually inspect the trajectories in 3D using tools such as the popular VMD [13], PyMOL [14], and chimera [15] (among others). While these run locally, other tools like MDsrv [16] or Mol* [17] have recently been put forward to conduct and share 3D MD analysis remotely via web-browser. However, visual inspection is not generally scalable beyond a certain number of trajectories or a certain number of atoms and is often not sufficient to identify or characterize key events.

Hence, very often, the next level of analysis will be offered by these same programs, either via GUI-menus and plugins or programmatically through scripting. Offered are general, community accepted metrics such as root-mean-square-deviation (RMSD), root-mean-square-fluctuation (RMSF), Ramachandran-plots [18], contact-maps, order-parameters, or more specific, user selected geometric values (distances, bond-angles, dihedral angles etc), or interaction types (Hydrogen bonds, salt-bridges, pi-stacking etc). Once arrived at the scripting/programmatic level, tools do not necessarily require a GUI, and can be used, even remotely, directly on the platform where the MD data resides. A very popular example are the analysis tools shipped with the GROMACS MD simulation suite [19], but many other standalone command-line tools provide these (and similar) analysis solutions, e.g., the Get-Contacts [20] command-line-tool or the popular Python modules MDtraj [21], MDanalysis [22], and Pytraj [23,24]. Some web-platforms also offer codeless, one-shot analysis, like the specialized protein-ligand-interaction-profiler [25], where a wide-range of interaction types can be resolved at atomic level. All these tools offer a diverse catalogue of metrics, deliver atomic-level insights, and are available for non-programming experts willing to learn basic scripting.

Finally, data-driven solutions -automated to varying levels- can be considered the next level of analysis, ranging from geometric clustering, to general dimensionality reduction techniques, to more comprehensive, Physics-based modelling like Markov-State-Modelling (e.g., PyEMMA [26], MSMBuilder [27]). Provided that these models can be constructed with the available data, they offer a general representation of the MD data that is compact, captures the underlying model's physics, and is fully predictive.

However, of particular interest for this paper are tools published as Python modules, in particular those offering an application programming interface (i.e., a Python API) like e.g. MDtraj, MDanalysis. The API enables users to build and combine their analysis workflow with the growing universe of well-documented and well-maintained scientific Python modules [28] (and references therein). Importantly, users can also fully exploit the feature-rich Jupyter Notebook, Jupyter-Lab and Google-Colab computing environments, which have become a widely popular scientific result-sharing platform [29].

Considering all of the above, mdciao is introduced in this rich software landscape trying to add value by:

- Taking non-expert users from their MD-data to a set of compact, production-ready tables and figures in one single shot, while remaining highly customizable for expert users, preferably in one step.

- Working with minimal user input.

- Focusing on a transparent, transferable, and universal metric that is understandable by experts and non-experts alike: residue-residue contact-frequencies with hard cutoffs.

- Exploiting available consensus nomenclature for bulk selection, manipulation, annotation and comparison purposes.

- Placing special care on user-friendliness, documentation (inline and online) and tutorials.

- Allowing for local computation and representation, i.e., no need to upload data to external platforms.

- Providing expert users a fully-fledged API to integrate mdciao into their workflows without having to leave the Jupyter Notebook platform.

## Design and implementation

### Basic principle

At the core of mdciao lies the computation of residue-residue distances, for which we have implemented a modified version of the `mdtraj.compute_contacts` method of MDtraj. This modification allows mdciao to keep track of the atom-types involved in the interactions, e.g., sidechain-sidechain, sidechain-backbone and so on. Then, for any given residue pair (A, B), once the distance $d_{AB}$ is computed for every frame, $t$, of every $i$-$th$ trajectory, the contact-frequency for that pair (A,B) in that trajectory, $i$, denoted $f^i_{AB,\delta}$, is extracted using a distance cutoff, $\delta$, using the formula:

$$f^i_{AB,\delta} = \frac{\sum_{j=0}^{N^i_t} C_\delta \left( d^i_{AB} \left( t_j \right) \right)}{N^i_t}$$

where $i$ is the trajectory index and $N^i_t$ is the number of frames in the $i$-$th$ trajectory. $C_\delta$ is the contact function, which depends parametrically on the cutoff value $\delta$ and is defined as:

$$C_\delta \left( d_{AB} \right) = \begin{cases} 1 \; if \; d_{AB} \leq \delta \\ 0 \; if \; d_{AB} > \delta \end{cases}$$

The average global contact frequency, $F_{AB,\delta}$, is then computed over all $T$ trajectories as

$$F_{AB,\delta} = \frac{\sum_{i=1}^{T} \sum_{j=0}^{N^i_t} C_\delta \left( d^i_{AB} \left( t_j \right) \right)}{\sum_{i=1}^{T} N^i_t}$$

and the individual per-trajectory frequencies, $f^i_{AB,\delta}$, as

$$f^i_{AB,\delta} = \frac{\sum_{j=0}^{N^i_t} C_\delta \left( d^i_{AB} \left( t_j \right) \right)}{N^i_t}$$

Given that the average, global frequency -over $T$ trajectories- might mask outlier, per-trajectory frequencies, mdciao makes both $F_{AB,\delta}$ and $f^i_{AB,\delta}$ values available for inspection.

By default, the cutoff $\delta$ is set at 4.5 Å, and the distances are computed between the closest heavy-atoms of two residues. This so-called distance "scheme", together with this cutoff value, in particular, has been chosen as a good trade-off between capturing short-range interactions, peaking at different distances (polar, non-polar, charged, and aromatic, among others) while introducing only one parametric dependency in the results, which are reported simply as "contacts". We have illustrated the relationship of distance distributions, distance cutoffs and contact frequencies for sample MD datasets [5,30–32], see S2, S3 and S4 Figs.

Users can, however, choose other cutoffs, in particular in combination with other distance schemes, like closest Cα-atoms, closest atoms (overall or just heavy atoms) and closest sidechain atoms (overall or just heavy atoms), all of which are implemented in mdtraj natively. Additionally, mdciao offers a "center of mass" scheme. These schemes can be passed as the "scheme" argument of all top-level, distance-computing methods of mdciao, which we discuss below.

## Basic Design Idea

We offer an overview of mdciao's structure and input/output workflow, together with a selection of plots in Fig 1. Following the object-oriented philosophy of other APIs such as mdtraj and mdanalysis, which encapsulate trajectory data into Trajectories and Universes, respectively, the basic design idea of mdciao is to encapsulate all distance-related data resulting from an MD simulation setup, that is, an arbitrary number of trajectories sharing a molecular topology and simulation parameters, into one single class, called `ContactGroup`. This object is automatically instantiated in the background, from user input, when using the command-line-interface (CLI) or when using the top-level API methods from `mdciao.cli.interface`, `residue_neighborhoods`, and `sites`. This object contains, as core data, the geometric distances between residues that have been computed by the modified version of `mdtraj.compute_contacts`, and additionally, other data such as: residue- and atom-pairs involved in each contact (per frame), molecular topology, molecular fragments (user defined or inferred automatically), fragment labels (user defined or inferred automatically), consensus labels of the residues, number of trajectories, individual trajectory length, trajectory filenames and more. This object, is generated by the functions downstream of these methods as stand-alone entity, and offers a variety of pre-packaged methods to easily generate frequency reports and plots, distance-plots such as time-traces or distributions, contact-matrices, flareplots and more. Beyond this, the encapsulation offers the advantage of serializing to a single ".npy" file for storage and re-use, as well as easy manipulation and comparison of `ContactGroup` objects for comparison of frequencies across different MD setups, regardless of the shape of the underlying MD dataset.

## Input and output

The needed minimal user input consists of:

- the residues or molecular fragments of interest, such as a ligand, a mutated site etc., two interfacing proteins or subdomains, or arbitrary groups of residues, expressed as generally as possible.

- the trajectory files to be analyzed. We note that we call these "trajectory files" because that is the most frequent case, but any ensemble of structures (e.g., members of a cluster, docking poses, or synthetic ensembles) stored in one or more files can be analyzed.

From this point on, with one command, mdciao automates all fragmentation, labeling, disambiguation, plotting and saving to file. Beyond this minimal input, the user may specify many other options, either by adding flags to the CLT or by using it in API mode to test different options live.

While mdciao is running, there is a live output informing of the different steps taking place. It becomes interactive if user-input is needed, e.g., for disambiguating two equally named residues, and finally produces text reports containing contact frequencies.

## Fragmentation heuristics

mdciao implements various heuristics to automatically split the molecular topology into different fragments. These heuristics are independent of the `chain` field of the PDB format,

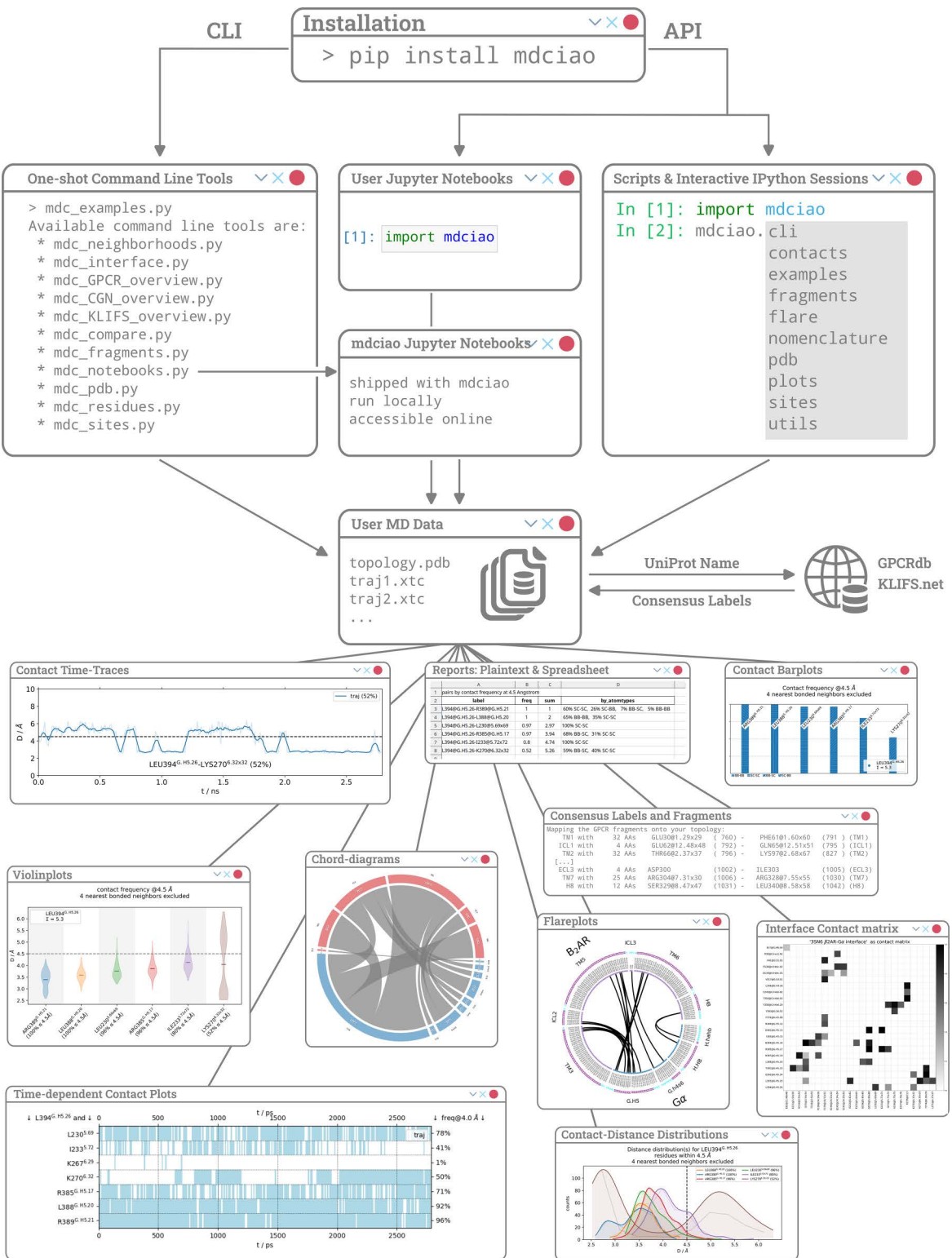

**Fig 1. Things to do after installing mdciao.** Users can choose the Command-Line-Interface (CLI, left branch) or the Application-Programming-Interface (API, right) to access mdciao's functionality. In the CLI, we recommend starting with `mdc_examples.py`, which offers pre-packaged examples that run locally out-of-the-box (see Table 1 for an overview of other commands and Table 2 for an overview of modules). In API mode, mdciao can be included in static scripts, interactive IPython sessions or Jupyter notebooks. Additionally, mdciao ships nine Jupyter notebooks that run sandboxed and persist with the installation.

which might not always be correct or be even present, as is the case in the popular.gro-file format. These heuristics use factors such as sequence jumps, presence or absence of bonds, residue names (protein vs non-protein, ion, water) to infer the underlying molecular topology. The so-recovered fragments group residues in meaningful ways, greatly simplifying both the user input and the annotated program output. Examples of the fragmentation heuristic being used can be seen in Figs 2, 3 (cell[4]), 4A, and in 5 (cell[4]).

### Consensus labeling: selection, annotation and alignment

Whenever possible, consensus nomenclature will be used to interpret the input and to annotate the output in texts, tables, and graphics. Currently, the implemented nomenclature databases are the GPCRdb [33] for GPCRs, the Common G-alpha Numbering (CGN [34]) for G-proteins and the KLIFS [35–37] for kinases. The user indicates the entry name via UniProt [38] names or UniProt accession codes，using either command-line flags, e.g., `--GPCR_UniProt adrb2_human` for the CLTs，or as API optional arguments, e.g., `CGN_UniProt='gnas2_human`. These codes are used to download the consensus nomenclature labels on-the-fly from their respective online databases or, alternatively, to read local files (Excel or plaintext files) which mdciao is able to generate and store for offline use (see Table 1). Subsequently, mdciao maps these labels via pairwise sequence alignment [39] and "tags" residues everywhere in the output with those labels. For example, in the residue-pair `R131@3.50-Y391@G.H5.23`, an extra bit of information is added succinctly: namely that, in the receptor, R131 is on helix 3, position 50 ([40]) and, on the G-protein, Y391 is on helix 5, position 23 ([34]). Additionally, consensus fragments are automatically inferred and labeled, s.t. mdciao will be aware of exactly what residues (and importantly, what indices) are contained in `TM6` (transmembrane helix 6 for a GPCR) or `G.H5` (helix 5 for a G-protein). These definitions can then, in turn, be used to quickly define interfaces of interest, e.g., for GPCR-- G-protein or GPCR—ligand interface. For example, specifying `ICL*` and `G.H*` will compute all contacts between intracellular loops (ICL1, ICL2, ICL3) with the Ras-Homology-domain of the G-protein without the user having to define them specifically. This is particularly useful when repeating the same computation for different (but related) systems, where residue indices might have changed, and off-by-one errors are likely to happen. In particular the notebook 09. Comparing Frequencies: Consensus Nomenclature exploits this selection syntax to compute TM3 contacts across four receptors with less than 30% sequence identity, coming from very different sources, without having to identify any particular residue by name or by index, but using simple selection syntax.

Finally, a direct consequence mapping consensus labels onto arbitrary user topologies in the same notebook is having access to a de-facto multiple-sequence alignment, which in turn allows for on-the-fly sequence-based structural alignments, as shown in the notebook 06. Use Consensus Labels as Multiple-Sequence-Alignment (MSA).

### API

The Application Programming Interface (API, Figs 1, 3 and 5) expands the functionalities of the command line tools (CLTs) and gives the more experienced users programmatic control of mdciao, allowing for the easy inclusion of its methods and classes into arbitrary Python workflows, via `import mdciao`. Crucially, any other, arbitrary Python modules that any user considers of importance for the problem at hand (clustering, time-series analysis, statistical modelling, plotting, formatting) can be used on mdciao's results without forcing the user to abandon the familiar and powerful (I)Python console or the Jupyter Notebook (see below). This is particularly useful, e.g., in Fig 5, where the 3D representation of relevant contacts is

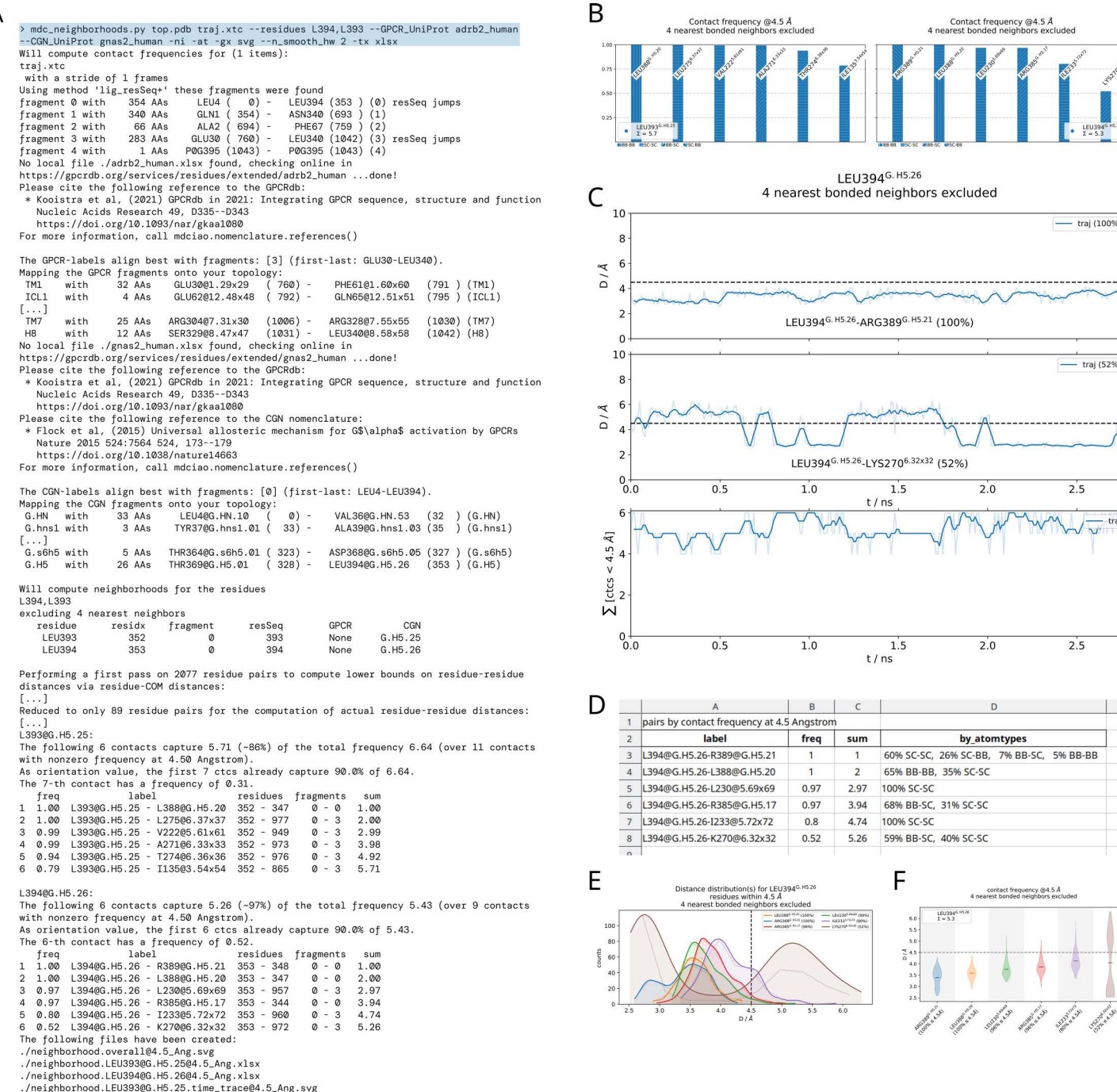

**Fig 2. Overview of one** `mdc_neighborhoods.py` **call from the command line.** A) Terminal input (on top, shaded) and output, with slight edits denoted as […]. The chosen residues are the C-terminal residues of the Gα-5 helix of the Gαβγ-protein, LEU393 and LEU394. B) Contents of the file `neighborhood.overall@4.5_Ang.svg`, showing the contact-frequencies represented as bars, which themselves contain information about interaction types (sidechain or backbone) encoded in their different hatching (i.e., the patterns filling the individual bars). Note the consensus labels, which help distinguish between G-protein residues and receptor residues. C) Contents of `neighborhoodLEU394@G.H5.26.time_trace@4.5_Ang.svg`, showing the smoothed time-traces of the residue-residue distances yielding the bars in panel A), also annotated with consensus labels and frequency values (three sub-panels have been edited out). The bottom panel of C) also contains the time-trace of the sum over all formed contacts, which oscillates around 5.3 as reported in A) and B). D) Snapshot of the `neighborhood.LEU394@G.H5.26@4.5_Ang.xlsx` spreadsheet containing the L394@G.H5.26 neighborhood, numerically specifying the interaction types hatched into the frequency bars of B). E) Alternative neighborhood representation, using residue-residue distance-distributions, providing more insight beyond the plain frequency values. F) Alternative neighborhood representation using violins. This allows for a representation as compact as panel B) but as informative as panel E). A full version of the text output can be found in the S1 Fig, and an online at https://proteinformatics.uni-leipzig.de/mdciao/overview.html. Locally, mdciao users can access this CLT example (and others) by invoking the CLT `mdc_examples.py` (cf. Table 1).

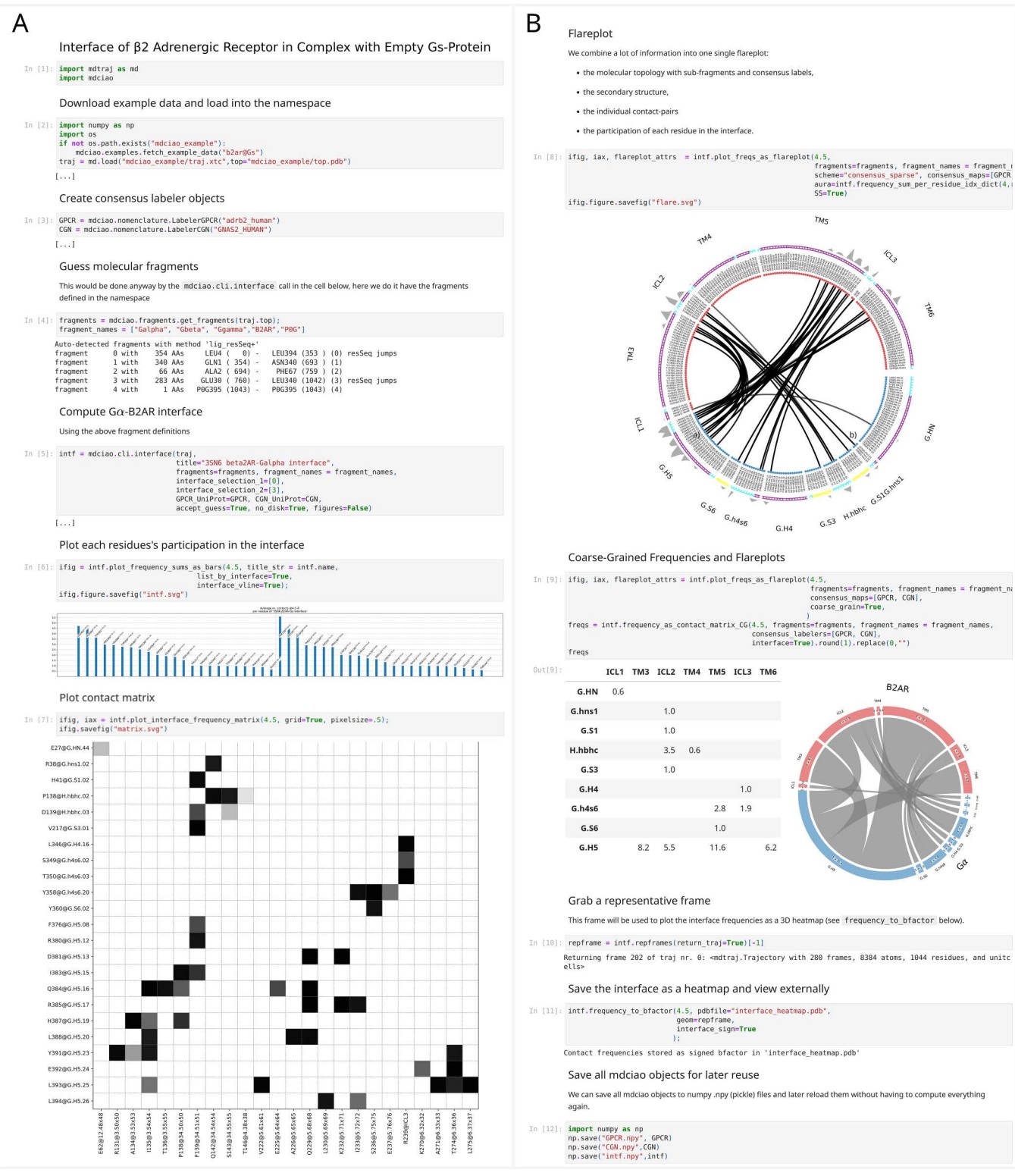

**Fig 3. Example Jupyter notebook illustrating how mdciao can be used in API mode.** In this case we compute the Gα-β2AR-interface contact frequencies. The 12 notebook cells are shown in two panels next to each other, A) cells [1] to [7] and B) cells [8] to [12]. The outputs of cells [3] and [5] have been edited out ([…]) but are analogous to Fig 2A) and can be found in the online documentation and the S1 Notebook. The main computation is the generation of the mdciao object, intf, in cell [5] which can later be used to generate multiple text, tabular, and graphic reports of the frequencies, distributions, and time-traces anywhere else in the notebook. A full version of this notebook, with full outputs and high-res pictures can be found in the S1 Notebook, and an online version can be found at https://proteinformatics.uni-leipzig.de/mdciao/gallery.html#examples. Locally, mdciao users can access this notebook (and others) by invoking the CLT mdc_notebooks.py (cf. Table 1).

**Table 1. Overview of command-line tools (CLTs) shipped with mdciao. These tools are one-shot tools that take users from basic input to production-ready figures and tables.**

| tool type | Command-line Tool (CLT) | Comment |
|---|---|---|
| learning tools | `mdc_notebooks.py` | create a fresh, ready-to-execute local copy of worked-out example Jupyter Notebook Tutorials ready to be run. They include the ones in this publication and others. |
| | `mdc_examples.py` | ready-to-execute, pre-filled examples for all the CLTs (add -x to execute) |
| pre-run tools | `mdc_pdb.py` | fetch and download RCSB PDB structure, including citation. |
| | `mdc_fragments.py` | overview of all available fragmentation schemes for a user-provided topology (e.g., a PDB file) |
| | `mdc_residues.py` | residue selection using flexible syntax, including consensus nomenclature, e.g., P0G,GLU*,380-394,3.5* |
| | `mdc_CGN_overview.py` | fetch (online or locally) consensus numbering labels. Produce an overview, e.g., of the fragments derived from the nomenclature. Optionally map the labels and fragments on a user provided topology. Optionally store the nomenclature locally. |
| | `mdc_GPCR_overview.py` | |
| | `mdc_KLIFS_overview.py` | |
| run tools based on residue-residue distances | `mdc_neighborhoods.py` | select residues with a very flexible input, e.g., `-r GLU*,GDP,L394,380-390` |
| | `mdc_interface.py` | select residues via fragments: automatically defined, user-specified, or derived from consensus nomenclature, e.g., TM5,TM6 |
| | `mdc_sites.py` | select by user specification of residue pairs of interest, e.g., `R135-E131, R135-E247` etc |
| post-run tools | `mdc_compare.py` | compare and combine results from different runs |

carried out "in-notebook" using `nglview` [41], and can thus be iteratively fine-tuned while having live access to the data.

The API is structured in submodules, geared mostly, but not exclusively, towards offering the CLT methods for programmatic use. Hence, the main module, `mdciao.cli`, provides direct ways to go from MD data to reporting and plotting contacts in one call. Other submodules (and submodules within) expose further methods, not necessarily conducive to contact-frequencies, but usable as standalone for purposes like consensus-labels retrieval and mapping, fragmentation heuristics, or plotting frequency values coming from different datasets. We have provided a summary of these modules and their intent in Table 2.

## Results

Mdciao's capabilities are offered via command-line tools (CLTs) and an API that can be imported to any (I)Python session (see Fig 1). As already outlined, the only input needed, beyond the MD trajectories, are residues or fragments of interest, and these can be specified directly or indirectly.

## Command-line tools

We present an overview of the command-line tools (CLTs) shipped with mdciao in Table 1. We have divided them into *pre-run*, *run*, and *post-run* CLTs, with two extra *learning* CLTs to help the user familiarize with mdciao. An example of the inputs and outputs of the CLT `mdc_neighborhoods. py` can be found in Fig 2. There, for a sample GPCR—G-protein system, we have chosen to compute the contacts of the two C-terminal residues of the Gα-5 helix of the Gαβγ-protein. Please note, this example can be run automatically by new users by issuing `mdc_examples.py neighborhoods` yielding the same results as in Fig 2. In panel A) we show the terminal output, with the results of the fragmentation heuristic recognizing the Gα, Gβ and Gγ subunits, the β2AR, and the receptor ligand PG0. The UniProt accession names `adrb2_human` and `gnas2_human` are used to retrieve GPCR and CGN nomenclature, respectively, from the GPCRdb [33]. The reference nomenclature is aligned to the user's input topology (which can be arbitrary) and exploited to map the consensus

**Table 2. Overview of mdciao's API submodule and their functionalities.**

| Submodule | Description |
|---|---|
| mdciao.cli | Programmatic access to the CLI methods |
| mdciao.contacts | Computation, bookkeeping, and manipulation of residue-residue contacts. |
| mdciao.fragments | Guess and manipulate fragments, i.e., sub-regions of molecular topologies. |
| mdciao.nomenclature | Get and manipulate consensus nomenclature for GPCRs, G-proteins, and Kinases. |
| mdciao.sites | Tools for reading and manipulating sites. |
| mdciao.plots | Plotting functions |
| mdciao.utils | Container for other sub-modules with lower-level functions. |
| mdciao.flare | Produce *flare-plots*, where the residues are drawn on a circle and connected with lines of varying opacity. |
| mdciao.pdb | RCSB-PDB web lookups |
| mdciao.examples | Helper functions for demos and data download |

fragments onto it. Finally, the neighbors and frequencies are reported as plaintext and listed to the output . The graphical outputs are shown in Fig 2B–2F.

## Example notebooks

Nine example Jupyter notebooks are shipped with mdciao, and they can always be accessed, run and modified locally in a *sandboxed* way by using the CLT mdc_notebooks.py (cf. Table 1). In Figs 3 and 5 we show two of these notebooks.

In the notebook shown in Fig 3, the goal is to compute the contacts of the Gα-β2AR-interface and represent them in different ways. Central to the notebook is the generation of an mdciao ContactGroup, using mdciao.cli.interface, in cell [5], which can later be used to generate multiple text, tabular, and graphic reports of the frequencies, distributions, and time-traces. In the notebook we show the per-residue interface-participation (cell [6]), the contact matrix (cell [7]), the flareplot (cell [8]) and its coarse-grained version (cell [9]). As can be seen also in Fig 4A), the flareplot can integrate many different types of information: the molecular topology with fragments (Gα, β2AR), the consensus subdomains (e.g., TM3 or G.H5), the contact frequencies of the individual residue pairs, the consensus labels of the residues, their secondary structure (letters C for coil, H for Helix and B for β-sheet) and an outer ring (aura-parameter in cell [8]), which can represent any per-residue numerical value. Here, we have plotted each residue's participation in the interface (i.e., the same bars as in cell [6]), but in principle any other (arbitrary) per-residue quantity could be included into the flareplot, e.g., the sequence conservation degree across a protein class, the root-mean-square-fluctuation (RMSF), the solvent-accessible-surface-area (SASA), the hydrophobicity, or any other informative numerical value that can be imported into the Python namespace. Back to Fig 3, mdciao selects, within the used MD trajectory data, a frame that is representative of the interface, upon which an interface heatmap is added as *bfactor* in cell [11]. Again, this amounts to inserting the bar-heights of cell [6] into the *bfactor* field of the PDB file inter-face_heatmap.pdb, which contains the representative frame. Note that the *bfactor* is signed (negative or positive) along interface definitions, meaning Gα-residues get negative bfactors and β2AR residues get positive bfactors. When reading the PDB-file into any 3D molecular viewer (e.g., in VMD with the blue-gray-red BGrR colormap, shown in Fig 4B), the signed *bfactor* highlights the molecular fragments in different colors, Thus, the 2D information of the flareplot can be readily identified, e.g., the middle of the G.H5 (blue) interacting with the tips of TM3, TM5, and ICL3 (red) or the ICL3 interacting lightly with the G.H4 and G.h4s6 subdomains of the Gα (in light blue).

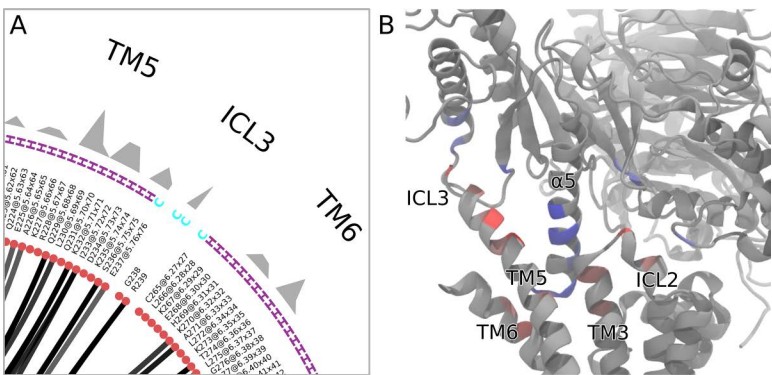

**Fig 4.** A) Zoomed-in section of Fig 3B). The flareplot after cell [8] shows the consensus labels of the residues, the secondary structure assignments (letters H or C for helical or coil, respectively) and the aura (gray ragged band) on the outer ring of the plot. In this case, it shows the sum of contact-frequencies for a given residue, tracking its participation in the interface (the same per-residue sum is represented in Fig 3A) cell [6]. Any per-residue scalar quantity can be represented this way, like SASA or RMSF among others. B) The residue participation, color-coded onto the representative frame (Fig 3B) cell [10]) using red for receptor residues and blue for G-protein residues.

In Fig 5 we show an example Jupyter notebook comparing ligand-kinase interactions for four different inhibitors bound to the Epidermal Growth Factor Receptor (EGFR). As in Figs 2 and 3, the main computation is the generation of the mdciao `ContactGroup`, using the method `mdciao.cli.interface`, for the four inhibitors EUX1, 7VH1, W321, and P31. As in Fig 3, we generate the nomenclature object on the fly, using the UniProt accession code P00533, which is associated to the kinase EGFR. Since the goal of the notebook is to compare datasets, we do not show individual plots as in Fig 3, but rather combine all the contact information into one compact violinplot, in cell [7] (please see the S2 Notebook for a large version of this picture). This plot shows (in vertical) the distributions of the residue-residue distances between the residues of the kinase binding pocket and the four inhibitors. The kinase residues are listed along the x-axis and are tagged with their KLIFS labels. In this plot, one can quickly appreciate individual differences in the binding patterns, in particular when more than one mode (per residue) is present. Furthermore, we use the `representatives` option to superimpose, on top of each violin, a single dot representing the residue-residue distance-value of the representative geometries, which are also returned to the namespace. These geometries are then optimally aligned (cell [9]) on the binding pocket (via the consensus KLIFS labels, see above) and are shown in [10] using `nglview` [41]. We highlight (in the same color as the violinplots in [7]) the kinase residues `C775@b.l.36`, P841@c.l.74, D855@xDFG.81, F997@EGFR, which all show different behavior for each inhibitor, as can be seen in the distributions of [7] and in the 3D visualization.

As a final note, whereas only a few (of many) use cases have been chosen for this manuscript, readers are highly encouraged to use mdciao's online tutorials and FAQs to get a full view of the software's capabilities. It should be noted that mdciao is not a GPCR-specific or kinase-specific tool and thus can be used with any system. Beyond our example notebook on the mutated interface of the SARS-CoV-2 spike protein receptor binding domain (RBD), bound to the human angiotensin converting enzyme-related carboypeptidase (ACE2), a growing list of works have used mdciao for other systems like viroporins [42], cyclooxygenase enzymes [43], or the Respiratory complex I [44].

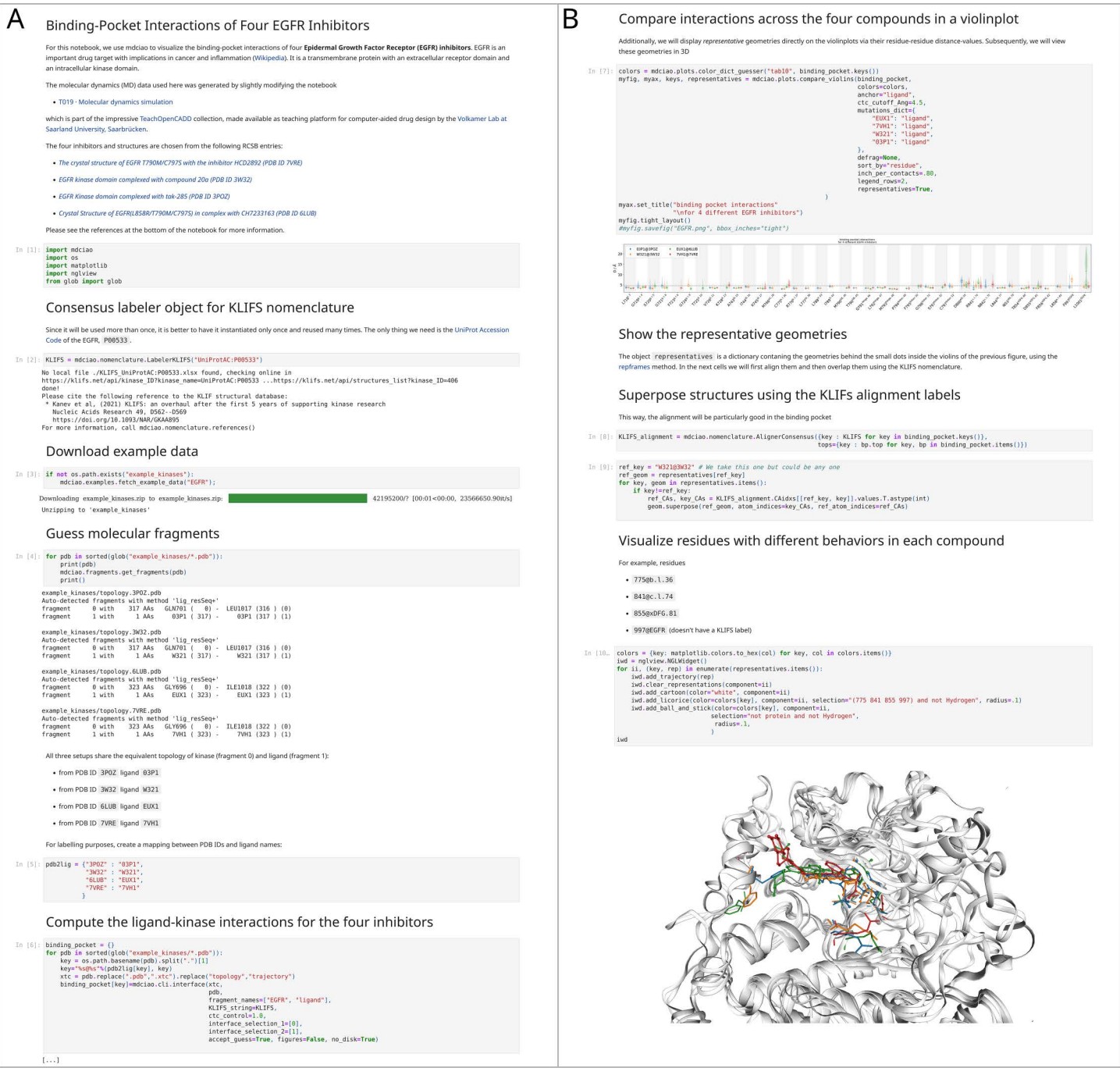

**Fig 5. Example Jupyter notebook illustrating how mdciao is used in API mode.** In this case we compute and compare ligand-kinase contacts for four different inhibitors bound to the Epidermal Growth Factor Receptor (EGFR). The 10 notebook cells are shown in two panels next to each other, A) cells [1] to [6] and B) cells [7] to [10]. The outputs of cell [6] have been edited out ([…]) but are analogous to Fig 2A) and can be found in the online documentation and the S2 Notebook. The main computation is the generation of the mdciao object, `binding_pocket`, in cell [6], for all four inhibitors: EUX1, 7VH1, W321, and P31. We combine and compare all the contact information into one compact violinplot in cell [7] (note the use of the KLIFS labels, please see the S2 Notebook for a large version of this picture). A full version of this notebook, with full outputs and high-res pictures can be found in the S2 Notebook, and an online version can be found at https://proteinformatics.uni-leipzig.de/mdciao/gallery.html#examples. Locally, mdciao users can access this notebook (and others) by invoking the CLT `mdc_notebooks.py` (cf. Table 1).

## Discussion

We present a user-friendly command-line tool that produces *one-shot* contact-frequency reports that are production-ready. It can be incorporated in any Python workflow via its API, and while it analyses MD data locally, it can contact online databases for rich annotation of the results. A variety of plotting functionalities have been implemented to quickly gain insight into the salient features of any MD dataset with little prior knowledge about the system, in particular for the computation of interfaces between bulk molecular fragments. Furthermore, for systems in which consensus nomenclature exists, analysis across different species, with very different primary sequences, can be streamlined without re-writing any code, for an arbitrary number of systems, a feature not presently implemented in any other software.

Considerable effort has been invested in making mdciao user-friendly. Firstly, it installs directly with the widely used `pip` Python manager via `pip install mdciao`. Secondly, `mdc_examples.py` offers new users a catalogue of ready-to-run, pre-packaged CLT-calls that use sample MD data already downloaded at installation. Furthermore, the documentation is extensive and accessible both inline (via the terminal, any integrated development environment (IDE), or the Jupyter Notebook) and online at https://www.mdciao.org. There, multiple FAQs, walkthroughs, and Jupyter Notebook Tutorials are presented to showcase most of mdciao's methods and present potential caveats. Additionally, these notebooks can always be accessed and modified locally in a *sandboxed* way by using the CLT `mdc_notebooks.py` (cf. Table 1).

## Limitations

Using a hard distance cutoff can over- or underrepresent some residue-residue interactions, since not all of these occur at the same residue-residue distance and relative position, e.g., salt-bridges vs. pi-stacking. vs. Hydrogen bonds. Other analysis tools like [13,20,22,25,45], use individual geometric definitions (distances and angles) for each interaction type. Their results then depend on each of those individual definitions, which may be more or less established, and more or less consistent across packages and publications. These have the obvious advantage of differentiating between interaction types, but also of resolving (in some packages and commercial suites) to the level of the atomic pair. This is particularly useful with non-peptidic ligands, where the notion of 'residue' does not apply properly and a single ligand can have several moieties, e.g., multiple aromatic rings in a drug-like molecule or different phosphate groups in a nucleotide.

However, with simplicity and transferability in mind, mdciao aggregates interactions to residues and depends parametrically only on one value (the hard cutoff) which is transparently presented in all of the reports. Ultimately, mdciao's analysis power relies first and foremost on differentiating between frequent and infrequent neighbors, and not on slight numerical frequency variations, which will systematically increase or decrease with a given cutoff.

## Availability and future directions

mdciao is published under the GNU Lesser General Public License v3.0 or later. The source code is hosted on https://github.com/gph82/mdciao, the current stable release is hosted at https://pypi.org/project/mdciao/ and the documentation, including guides and examples, can be found at https://www.mdciao.org. The release used for this manuscript is v.1.0.0.

Community contributions are welcome, and we are committed to provide continuous support through https://github.com/gph82/mdciao/issues.

The entire inputs and outputs of the mdciao calls presented in Figs 2–3 and 5 can be found in the S1 Figure, and the S1 and S2 Notebooks, respectively.

## Supporting information

**S1 Fig. Complete text output printed to the terminal by the CLT (shown in Fig 2A) of the main text.**
(TIF)

**S2 Fig. Distribution of closest heavy-atom—heavy-atom distances over five different MD datasets (S1 Table).** The distributions are shown for all residue-pair types in a), and then for specific residue types: b) residues with charged sidechains, c) residues with aromatic side-chains, discarding the polar OH-group in PHE and d) residues with nonpolar sidechains. These simulations have been carried out with different (but comparable) simulation engines, parameters and forcefields, but roughly recapture each other's peak-positions, with slight shifts along the x-axis. Selected, longer-range peaks denoted with *, ‡, †, ‡, and ⅃ are shown in **S4 Fig** via snapshots of the trajectory.
(TIF)

**S3 Fig. Correlation (a) and Mean-Absolute-Error (b) between frequencies $F_{AB,\delta_0}$ at a reference cutoff, $\delta_0$, and different frequencies, $F_{AB,\delta_{scan}}$, at different cutoff-values, $\delta_{scan} \in [3.0, 5.5]$ Å, for selected, representative peaks (c,d,e) of the heavy-atom—heavy-atom distance-distributions shown in S2 Fig.** The peaks have different centers and shapes depending on the interaction type and on the dataset used, hence for this sample we have chosen representative distributions in which only one peak is captured (gray area under the curves). Panel a) shows the high (>.95) correlations around $\delta_{scan}$ between 4-5 Å. We have included the default value of mdciao $\delta = 4.5$ Å as a dotted vertical line. Panel b) shows the mean-absolute-error (MAE) in the frequencies, in absolute percentage terms. Around 4.5Å, the MAE is less than 5% (solid lines) and standard deviation is around that value (the 95% confidence is shown as shaded area).
(TIF)

**S4 Fig. Longer-range peaks of S2 Fig (using arrows and the *, ‡, †, ‡, and ⅃ symbols) not corresponding to short-range contacts, but rather secondary structure elements like beta-sheets or alpha helices.** Residue pairs like the ones shown in panels a), b), c) and d) are in the vicinity of each other by virtue of sharing beta-sheet structure, but are offset with respect to the actual backbone-backbone hydrogen-bond interaction. e) denotes residue pairs corresponding to the fifth residue after a full alpha-helical turn.

(TIF)

**S1 Table. Overview of the MD datasets used in S2 Fig.**
(XLSX)

**S2 Table. Walltimes for the contact-frequency data for the MD datasets used in S2 Fig on a desktop computer, Intel Core i7-6700K CPU @ 4.00GHz, with 8 cores and 64GB RAM.** The files were streamed from disk in blocks of 500 frames.
(XLSX)

**S1 Notebook. Full inputs and outputs of Fig 3 in high resolution.**
(PDF)

**S2 Notebook. Full inputs and outputs of Fig 5 in high resolution.**
(PDF)

## Acknowledgments

G.P.H. thanks Hossein Batebi and Ramon Guixà-González for their valuable comments and role as beta testers. P.W.H and G.P.H thank Sofi Tiwari for her initial contributions to some precursor utilities of mdciao.

## Author contributions

**Conceptualization:** Guillermo Pérez-Hernández.

**Data curation:** Guillermo Pérez-Hernández.

**Formal analysis:** Guillermo Pérez-Hernández, Peter W. Hildebrand.

**Funding acquisition:** Peter W. Hildebrand.

**Investigation:** Guillermo Pérez-Hernández, Peter W. Hildebrand.

**Methodology:** Guillermo Pérez-Hernández.

**Project administration:** Guillermo Pérez-Hernández, Peter W. Hildebrand.

**Resources:** Peter W. Hildebrand.

**Software:** Guillermo Pérez-Hernández.

**Supervision:** Peter W. Hildebrand.

**Validation:** Guillermo Pérez-Hernández, Peter W. Hildebrand.

**Visualization:** Guillermo Pérez-Hernández, Peter W. Hildebrand.

**Writing – original draft:** Guillermo Pérez-Hernández, Peter W. Hildebrand.

**Writing – review & editing:** Guillermo Pérez-Hernández, Peter W. Hildebrand.

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
