## [Decision Letter · Decision Letter 0]

25 Nov 2024

PCOMPBIOL-D-24-01810mdciao: Accessible Analysis and Visualization of Molecular Dynamics Simulation DataPLOS Computational Biology Dear Dr. Pérez-Hernández, Thank you for submitting your manuscript to PLOS Computational Biology. After careful consideration, we feel that it has merit but does not fully meet PLOS Computational Biology's publication criteria as it currently stands. Therefore, we invite you to submit a revised version of the manuscript that addresses the points raised during the review process. Please submit your revised manuscript within 60 days Jan 25 2025 11:59PM. If you will need more time than this to complete your revisions, please reply to this message or contact the journal office at ploscompbiol@plos.org. Please include the following items when submitting your revised manuscript: * A rebuttal letter that responds to each point raised by the editor and reviewer(s). You should upload this letter as a separate file labeled 'Response to Reviewers'. This file does not need to include responses to formatting updates and technical items listed in the 'Journal Requirements' section below.* A marked-up copy of your manuscript that highlights changes made to the original version. You should upload this as a separate file labeled 'Revised Manuscript with Track Changes'.* An unmarked version of your revised paper without tracked changes. You should upload this as a separate file labeled 'Manuscript'. If you would like to make changes to your financial disclosure, competing interests statement, or data availability statement, please make these updates within the submission form at the time of resubmission. Guidelines for resubmitting your figure files are available below the reviewer comments at the end of this letter. We look forward to receiving your revised manuscript. Kind regards, Alexander MacKerellAcademic EditorPLOS Computational Biology Arne ElofssonSection EditorPLOS Computational Biology Feilim Mac GabhannEditor-in-ChiefPLOS Computational Biology Jason PapinEditor-in-ChiefPLOS Computational Biology  **Journal Requirements:**

2) Please provide an Author Summary. This should appear in your manuscript between the Abstract (if applicable) and the Introduction, and should be 150u2013200 words long. The aim should be to make your findings accessible to a wide audience that includes both scientists and non-scientists. Sample summaries can be found on our website under Submission Guidelines:

3) Your manuscript is missing the following sections: Design and Implementation, and Availability and Future Directions. Please ensure that your article adheres to the standard Software article layout and order of Abstract, Introduction, Design and Implementation, Results, and Availability and Future Directions. For details on what each section should contain, see our Software article guidelines:

https://journals.plos.org/ploscompbiol/s/submission-guidelines#loc-software-submissions

5) We have noticed that you have uploaded Supporting Information files, but you have not included a list of legends. Please add a full list of legends for your Supporting Information files after the references list.

Potential Copyright Issues:

- Figure 1. Please confirm whether you drew the images / clip-art within the figure panels by hand. If you did not draw the images, please provide (a) a link to the source of the images or icons and their license / terms of use; or (b) written permission from the copyright holder to publish the images or icons under our CC BY 4.0 license. Alternatively, you may replace the images with open source alternatives. See these open source resources you may use to replace images / clip-art:

7) Please amend your detailed Financial Disclosure statement. This is published with the article. It must therefore be completed in full sentences and contain the exact wording you wish to be published. Please ensure that the funders and grant numbers match between the Financial Disclosure field and the Funding Information tab in your submission form. Note that the funders must be provided in the same order in both places as well.

 **Reviewers' comments:** Reviewer's Responses to Questions

**Comments to the Authors:**

Reviewer #1: review is uploaded as an attachment

Reviewer #2: Pérez-Hernández and Hildebrand report an elegant ethod to conveniently analyze Md trajectories with particular focus on inter-residue (or residue-ligand) contact mapping. It delivers a number of very useful, high resolution plots, allowing easily to compare systems (i.e. with different ligands, with mutations), particularly-but not limited to- on the cases of GPCRs, G-proteins ir kinases, as it adapts the standard nomenclature on those cases. A limitation that is conveniently outlined by the authors is the lack of decomposing on atom-residue mapping, probably more suited to drug-design projects, but the scope of mdciao is clearly stated to be on the protein-protein interactions (inter or intraprotein) and as such it is a very complete method.

The software installs easily, it is also available on-line, very well documented with a number of examples of e-g- jupyter-notebooks that can be used as a starting point for novel users.

I have only a few minor corrections of typos or grammar as follows:

- line 41- "MDanalysis, The API" replace comma by dot

- line 153 - "a growing list works " probably "a growing list OF works (studies?)"

Reviewer #3: The manuscript “mdciao: Accessible Analysis and Visualization of Molecular Dynamics Simulation Data” by Perez-Hernandez and Hildebrand presents a software, actually command line tools and a python API, for analysis and representation of molecular dynamics (MD) simulation data. While many programmes exist to analyse and/or visualise MD data, including python modules rich in functionality, the authors claim to fill a gap by catering to non-expert users (with production-ready tables and figures) and expert users alike (by full flexibility to use the API in their python workflow).

The software is indeed useful and the description and examples provided in the present manuscript provide a good overview and introduction into its functionality, in particular thanks to the access to worked examples.

However, a few things need to be improved in the manuscript:

In the list of the added values, the authors mention, among others,

“residue-residue contact-frequencies with hard cutoffs” as an easy and universal metric.

However, it is not fully explained why this is desirable as the most important , if not only (?) metric, in the first place.

The authors discuss that such contacts defined via a hard cut-off are preferable over more fine-tuned parameters and should therefore be transferable. Given the importance of the contacts and the corresponding cut-off, it should be detailed more in the methods section of the manuscript. That is which of the “typically 3.5-4.5A” is chosen (say, as user-friendly default)? To which atoms does this distance cutoff apply? To all atoms of two residues in question? Only heavy atoms? The nearest two?

Furthermore, the authors claim to “special care on user-friendliness, documentation (inline and online) and tutorials”. The first can be tested by only using the software – and appears to be achieved for this reviewer. Documentation and tutorials are provided as example notebooks, or learning command line tools, and these are highly appreciated.

The link to documentation and tutorials should be placed more prominently in the manuscript, perhaps even in the abstract close to the link to the github repository.

**Have the authors made all data and (if applicable) computational code underlying the findings in their manuscript fully available?**

Reviewer #1: Yes

Reviewer #2: Yes

Reviewer #3: Yes

PLOS authors have the option to publish the peer review history of their article (what does this mean? ). If published, this will include your full peer review and any attached files.

**Do you want your identity to be public for this peer review?** For information about this choice, including consent withdrawal, please see our Privacy Policy .

Reviewer #1: **Yes: ** Adithya Polasa

Reviewer #2: **Yes: ** Hugo Gutiérrez-de-Terán

Reviewer #3: No

 **Figure resubmission:**While revising your submission, please upload your figure files to the Preflight Analysis and Conversion Engine (PACE) digital diagnostic tool, https://pacev2.apexcovantage.com/. PACE helps ensure that figures meet PLOS requirements. To use PACE, you must first register as a user. Registration is free. Then, login and navigate to the UPLOAD tab, where you will find detailed instructions on how to use the tool. If you encounter any issues or have any questions when using PACE, please email PLOS at figures@plos.org. Please note that Supporting Information files do not need this step. If there are other versions of figure files still present in your submission file inventory at resubmission, please replace them with the PACE-processed versions. 
---

## [Editor Report · Decision Letter 1]

31 Jan 2025

Dear Dr. Pérez-Hernández,

We are pleased to inform you that your manuscript 'mdciao: Accessible Analysis and Visualization of Molecular Dynamics Simulation Data' has been provisionally accepted for publication in PLOS Computational Biology.

Best regards,

Alexander MacKerell

Academic Editor

PLOS Computational Biology

Arne Elofsson

Section Editor

PLOS Computational Biology

---

## [Editor Report · Acceptance letter]

PCOMPBIOL-D-24-01810R1

mdciao: Accessible Analysis and Visualization of Molecular Dynamics Simulation Data

Dear Dr Pérez-Hernández,

I am pleased to inform you that your manuscript has been formally accepted for publication in PLOS Computational Biology. Your manuscript is now with our production department and you will be notified of the publication date in due course.

With kind regards,

Lilla Horvath
